qPCR multiplex detection of microRNA and messenger RNA in a single reaction

Khoury Samantha 1
Tran Nham nham.tran@uts.edu.au 2 3
1 Office of the Deputy Vice Chancellor Innovation and Enterprise, University of Technology Sydney , Sydney , New South Wales , Australia
2 Centre for Health Technologies and School of Biomedical Engineering, University of Technology Sydney , Sydney , New South Wales , Australia
3 The Sydney Head and Neck Cancer Institute, Royal Prince Alfred Hospital , Sydney , New South Wales , Australia
Sotelo-Mundo Rogerio
Electronic publication date: 2020 Jun 25
Publication date: 2020
Volume: 8
Electronic Location ID: e9004
Received 2020 Jan 6; Accepted 2020 Mar 26
Copyright: ©2020 Khoury and Tran
Copyright year: 2020
Copyright holder: Khoury and Tran
License: This is an open access article distributed under the terms of the Creative Commons Attribution License, which permits unrestricted use, distribution, reproduction and adaptation in any medium and for any purpose provided that it is properly attributed. For attribution, the original author(s), title, publication source (PeerJ) and either DOI or URL of the article must be cited.
License URL: https://creativecommons.org/licenses/by/4.0/

Keywords: qPCR multiplex detection of multiple miRNAs in serum, Low volume qPCR detection

Funding: Australian Federation of Graduate Women Barbara Hale Fellowship Samantha Khoury was supported by the Australian Federation of Graduate Women Barbara Hale Fellowship. The funders had no role in study design, data collection and analysis, decision to publish, or preparation of the manuscript.

==============================
Reverse Transcription-Quantitative PCR (RT-qPCR) is one of the standards for analytical measurement of different RNA species in biological models. However, current Reverse Transcription (RT) based priming strategies are unable to synthesize differing RNAs and ncRNAs especially miRNAs, within a single tube. We present a new methodology, referred to as RNAmp, that measures in parallel miRNA and mRNA expression. We demonstrate this in various cell lines, then evaluate clinical utility by quantifying several miRNAs and mRNA simultaneously in sera. PCR efficiency in RNAmp was estimated between 1.8 and 1.9 which is comparable to standard miRNA and random primer RT approaches. Furthermore, when using RNAmp to detect selected mRNA and miRNAs, the quantification cycle (Cq) was several cycles lower. This low volume single-tube duplex protocol reduces technical variation and reagent usage and is suitable for uniform analysis of single or multiple miRNAs and/or mRNAs within a single qPCR reaction.

Introduction

The advent of RNA sequencing (Mortazavi et al., 2008) and its application in both basic and clinical research has expanded our understanding of the human transcriptome. The RNA family extends beyond the messenger RNA (mRNA) and now includes long non-coding RNAs (lncRNAs), microRNAs (miRNA) (Lau et al., 2001; Lee & Ambros, 2001), circular RNAs (Memczak et al., 2013) and other newly discovered family members (Consortium, 2012). Smaller RNAs such as miRNAs are potent regulators of gene expression directing developmental pathways (Carrington & Ambros, 2003). MicroRNAs are touted as the next generation of biomarkers for different human diseases and biological states (Khoury & Tran, 2015). Consequently, the independent validation of specific RNA species either as gene regulators or clinical biomarkers requires a corresponding leap in the current quantitative detection technologies.

The most common method for detecting non-coding RNAs (ncRNA) and protein-coding messenger mRNAs is a two-step process referred to as Reverse Transcription-qPCR (RT-qPCR) (Bustin et al., 2005). A method universally practiced, RT-qPCR relies on Reverse Transcription (RT) and complementary DNA (cDNA) synthesis. This is followed by qPCR utilizing various probes or dyes for uniform application and fluorescent detection (VanGuilder, Vrana & Freeman, 2008). All steps are highly sensitive, requiring precision from the earliest stages of nucleic acid extraction through to the preparation of cDNA products and choice of reagents. A technical restriction of the current technology is the requirement for different single RT reactions amplifying one RNA subclass at a time. This restriction is due to different RNA species requiring certain cDNA priming conditions. For instance, small ncRNA such as miRNA, require specific stem loop primers or 3′ specific primers for cDNA amplification. The difference in primers contrasts to randomly selected primers utilized for mRNA transcripts (Chen et al., 2005; Li et al., 2014; Wang, Ach & Curry, 2007).

Consequently, most researchers will need to generate two separate RT reactions for miRNAs and mRNAs. Separate reactions double the cost, forming limitations on the amount of starting material and increasing time consumption when validating large gene sets. Hence, the “RNAmp” protocol was created, designed to simultaneously prime small ncRNA and mRNA transcripts within a single cDNA preparation which would then allow for the parallel detection.

Our technique is centered on Hydrolysis probes, one of the most common detection chemistries for RT-qPCR. RNAmp lends itself to utilizing multiple fluorescently labeled reporter genes to detect more than one RNA species within a single qPCR reaction. In order to achieve a broad detection between different RNA families, we modified the RT synthesis step combining random primers with stem loop primers. Furthermore, RNAmp also allows the user to subsequently detect both miRNA and mRNAs species in a single qPCR reaction. We compared RNAmp to current RT-qPCR standards demonstrating the advantages and utility of detecting both mRNA and miRNA species in cell lines, human serum and in a siRNA/miRNA knockdown system.

Material and Methods

Minimum information for publication of qPCR experiments

Compliance with the Minimum Information for the Publication of Real-Time Quantitative PCR Experiments (MIQE) guidelines (Bustin et al., 2009) has been provided and all experiments performed in triplicate.

RNA isolation and quantification of HEK 293, HeLa cell lines and human serum

HEK 293 (ATCC) and HeLa cell (ATCC) lines were grown to 90% confluency in DMEM, Glutamax (GIBCO) was supplemented with 10% FBS (GIBCO) and 1% Penicillin-Streptomycin-Glutamine (GIBCO) and total RNA was isolated using the Trizol method (Molecular Research Centre Inc). Total RNA from human serum was extracted and isolated using the Tri-reagent RT-LS protocol (Molecular Research Centre Inc.) (Khoury, Ajuyah & Tran, 2014). Serum samples were from cancer patients and healthy volunteers obtained from clinical collaborators at Royal Prince Alfred Hospital Sydney. The samples were stored at -80C till use. (Protocol number X10-0016 and HREC/10/RPAH/24). HREC was granted (UTS HREC 2013000471) by UTS Human Research Ethics Committee for use of the serum. Written consent for the human serum used in these experiments were provided by the patient utilising our HREC form. RNA quality and concentration were assessed with a Nanodrop 1000 (ThermosFisher) (See Table S1) and resuspended with RNase free dH20 to concentrations of 50 ng/µl, 100 ng/µl and 200 ng/µl for use in RT-qPCR.

miRNA mimic transfections in HeLa cell lines

HeLa cells were transfected with the Ambion Pre-miR miRNA Precursors (ThermosFisher) hsa-miR-21 mimic (20 pmol) (Invitrogen), as well as a Scramble #1 control (20 pmol) (ThermosFisher), in triplicate using Lipofectamine RNAiMAX Transfection Reagent (Thermo). Cells were seeded at a density of 4 × 105 into a 6 well plate and incubated for 24 h prior to transfection. All cells were collected 24 h post-transfection and total RNA extracted.

RT Priming strategies for synthesis of cDNA products

HEK 293 and HeLa total RNA were prepared using three different RT cycle conditions (See Table S2). MicroRNA and mRNA cDNA synthesis were performed according to the manufacturer’s protocol (Applied Biosystems) using stem loop and random primers respectively. RNAmp combines both primer sets to generate both mRNA and miRNA transcripts. All three RT approaches were generated with the High-capacity TaqMan miRNA Reverse Transcription Kit (ThermoFisher) using clear 0.5ml PCR grade tubes on an Eppendorf Mastercycler Pro S Vapoprotect.

For the first condition, we generated a 15.0 µl microRNA synthesis cDNA reaction. The included 4 U of 20 U/µL RNase inhibitor, a total volume of 6 µl miRNA RT primer mix, 50 U of MultiScribe™ Reverse Transcriptase, 50 Units/µL, 1× volume of 10× RT Buffer, 1 mM dNTP were combined and mixed with total RNA (See Tables S2 and S3). To generate the random prime cDNA reaction, 20 U of 20 U/µL RNase inhibitor, 1× of 10× RT Random primer, 50 U of MultiScribe™ Reverse Transcriptase 50 Units/µL, 1× of 10× RT Buffer and 4 mM dNTP were mixed and combined with total RNA (See Tables S2 and S3). Our third condition was the RNAmp which is a 15.0 µl mixture consisting of the following; 4 U of 20 U/µL RNase inhibitor, a total volume of 6 Textmu l miRNA RT primer mix, 50 U of MultiScribe™ Reverse Transcriptase, 50 Units/µL, 1× of 10× RT Buffer, supplemented with MgCl2 25 mM, 1.6 mM of 100 mM dNTP, and 1× of 10× RT Random primer (See Tables S2 and S3).

Singleplex and duplex hydrolysis based quantitative PCRs

Complementary DNA products from all three RT methods (micoRNA cDNA synthesis,mRNA cDNA synthesis and RNAmp) were diluted 1:4 with water and added to a Hydrolysis based Universal PCR master mix in four different volumes, 20 µl, 10 µl, 5 µl and 2.5 µl. Refer to Tables S4 and S5 for the composition of subsequent singleplex and duplex reactions. Hydrolysis probes were labelled as VIC; B2M, ACTB, GAPDH, 18S, hsa-miR-21 and FAM; P53, p16, JAG1, CDKN1, Dicer, Ago2, hsa-miR-99b, hsa-miR16, U75, JAG1, hsa-miR-486-5p, hsa-miR-451 and ordered from the manufacturer (Applied Biosystems). Singleplex reactions had a 1× volume of Taqman Universal PCR Master and 0.5 × (20.0 µl Final Volume), 1 × (15.0 µl Final Volume), 2 × (10.0 µl Final Volume), 4 × (5.0 µl Final Volume) and concentration of the target-specific TaqMan Real Time PCR primer probe set (Applied Biosystems). Those designated for duplex had a 1× concentration of Taqman Universal PCR Master for all except the 2.5 µl final volume reaction of 0.8×. The Taqman specific primer probe sets had 0.5× each (20.0 µl Final Volume), 1× each (15.0 µl Final Volume), 2× each (10.0 µl Final Volume), and 4× each (5.0 µl Final Volume) also from Applied Biosystems. From these master-mix solutions, triplicate qPCR reactions were carried out on an Applied Biosystems Step One with the following thermal-cycling procedure; 95 °C for 10 min, followed by 40 cycles of 95 °C for 15 s and 60 °C for 1 min (1.6 °C/s ramp rate) as specified by the manufacturer (Applied Biosystems). The experimental design for duplex qRT-PCR is more complicated than a single-plex. To avoid overlap of emission spectra we chose dyes with appropriate excitation wavelengths and little to no overlap in their emission spectra: VIC, FAM and ROX (Step One Plus passive reference dye). The settings for excitation and emission filters of real-time detection using Thermofisher reagents on a Step One Plus has been supplied in Table S7. The Step One Plus instrument was calibrated for each compatible dye; VIC, ROX and FAM as part of the experiment optimization process and per manufacturer recommendations. Appropriate calibration and choice of dye combinations enhance the dye specificity, and minimizes background and overlap of fluorescent signals.

LinRegPCR analysis of RT-qPCR data

Step One plus experimental files were exported and reformatted for LinRegPCR analysis and determination of Cq (Tuomi et al., 2010). Baselines were determined and amplicon groups were assigned with the following exclusion criteria; samples under analysis; samples without amplification, samples without plateau phase, samples with low Cq value and samples being outside of the 5% of the group median efficiency per amplicon. Furthermore, a log linear phase parameter during estimation of baseline was included. The qPCR efficiencies were exported and statistically analysed.

Statistical analysis

Mean values and efficiency for each Amplicon and reaction were calculated throughout with Standard Error of the Mean, Minimum, Maximum, Mean, and Standard Deviation. One-way ANOVA analysis was performed on multiple groups, to determine statistical significance. P values range from **** P < 0.0001, *** P < 0.001 and analysed using the Prism 6 software package.

Results

Low volume qPCR reactions improves PCR detection

First, we sought to improve the fluorescent based qPCR without the additional need of increasing RNA input and reagents. The standard TaqMan singleplex reaction volume was modified and compared between 20.0 µl, 10.0 µl, 5 µl to 2.5 µL (See Table S4). Two categories of RNA transcripts Reference genes and miRNAs, were quantified by Quantification cycle (Cq). The Cq is defined by the first detection of the amplicon above the RNA background and inversely correlated with abundance. For example, a lower Cq represents higher abundance of the original target RNA.

The results indicate the Cq value of Beta-2-Microglobulin (B2M) and Beta-Actin were decreased with smaller reaction volumes (Fig. 1A and Table S6). At the 2.5 µl reaction volume, the Cq value of Beta-Actin was 17, a difference of 9 Cq values from the 20 µl reaction. A similar trend of Cq was also observed for B2M. This change in the Cq threshold in decreasing reaction volumes was also observed for hsa-miR-21 and hsa-miR-99b (Fig. 1B and Table S6) with an average improvement of 5 Cq’s. These Cq shifts are equivalent to a 128- and 32-fold increase in detection for hsa-miR-21 and hsa-miR-99b, respectively. Furthermore, these changes in Cq values across the smaller volume groups for both B2M, Beta-Actin, hsa-miR-21 and hsa-miR-99b were statistically significant as determined by one-way ANOVA. Given the interest in using serum miRNAs as biomarkers, we tested if a commonly deregulated microRNA, hsa-miR-16, could be detected in human serum and improved by using smaller reaction volumes (Fig. 1C and Table S6). A similar outcome was observed at smaller reaction volumes.

Figure 1 Reduction in hydrolysis based qPCR reaction volumes lowers Quantification Cycle (Cq) values and increases detection sensitivity.

Note that on the Y axis, the Cq values are inverted and Cq values do not start from 0 to 40. Instead a selected Cq range was plotted to better visualize the shift in Cq values. Typically, a low Cq represents a higher sensitivity as the amplicon is detected at an earlier quantification cycle threshold. (A) Detection of reference genes Beta-Actin and B2M in qRT-PCR volumes of 20, 10, 5.0, and 2.5 µL. (B) Detection of hsa-miR-21 and hsa-miR-99b in 20, 10, 5.0, and 2.5 µL volumes. (C) Detection of miR-16 in human serum in reaction volumes of 20, 10, 5.0, and 2.5 µL. (D) Duplex detection of hsa-miR-21 and a reference gene U75. For each of the amplicons tested, there was a statistically significant difference between the different volume groups as determined by one-way ANOVA; B2M: P < 0.0001, Beta-Actin: P = 0.0008, hsa-miR-21: P < 0.0001, hsa-miR-99b: P < 0.0001. For the duplexing, hsa-miR-21: P = 0.0010 and U75: P = 0.0016.

Most miRNA qPCR hydrolysis probe-based assays are run as singleplex reactions. When detecting multiple miRNAs, this method increases reagent usage and is problematic when using limiting or rare samples. To alleviate this constraint, we evaluated duplexing for the detection of hsa-miR-21 and another RNA, U75 in working volumes of 10.0, 5.0, and 2.5 µL. U75 was selected, as it is a common small nucleolar RNA used as a reference gene for normalizing miRNA qPCR expression data (Livak & Schmittgen, 2001). As seen in Fig. 1D, the two targets were detected across the different reaction volumes. Moreover, these amplicons were detected at lower Cq values in smaller reaction volumes.

To eliminate any possibility of amplification bias at these lower volumes, we determined the qPCR efficiency using the software LinRegPCR (Ramakers et al., 2003; Tuomi et al., 2010). Using representative examples, B2M and hsa-miR-21, the qPCR efficiencies were similar in all the volumes tested (Table 1). Statistically there were no significant differences between the group means as determined by one-way ANOVA. Therefore, the reduction in reaction volumes does not impact on qPCR efficiency and PCR detection is directly dependent on smaller reaction volumes.

Table 1 PCR efficiency for B2M and miR-21 at different reaction volumes.

Reducing qRT-PCR reaction volumes does not affect PCR efficiency for the detection of these amplicons.

Volume	20.0 µL	10.0 µL	5.0 µL	2.5 µL	
PCR efficiency for B2M	
Minimum	1.6	1.7	1.6	1.8	
Maximum	1.7	1.8	1.8	2	
Mean	1.7	1.8	1.7	1.8	
Std. Deviation	0.075	0.036	0.095	0.126	
Std. Error of Mean	0.043	0.021	0.055	0.067	
PCR Efficiency for miR-21	
Minimum	1.6	1.7	1.7	1.6	
Maximum	1.8	1.7	1.8	2.0	
Mean	1.7	1.7	1.7	1.8	
Std. Deviation	0.070	0.019	0.110	0.269	
Std. Error of Mean	0.040	0.011	0.063	0.155	

We then investigated whether RNA concentration influences Cq values at these low volumes. Total RNA inputs of 50 ng, 100 ng and 200 ng were used to generate the two-standard manufacturer RT reactions for individual detection of RNA and miRNA species. Please refer to Table S1 for RNA Concentration and Quality. For the RNA species, Beta-Actin, GAPDH, 18s and p53 (Fig. 2A), a consistent Cq level was observed across these concentrations. Applying the manufacturer’s protocol to small RNAs (Fig. 2B) hsa-miR-21, hsa-miR-99b, U75 and Let-7b, the same result was obtained. Taken together, these results suggest qPCR detection at low reaction volumes does not solely depend on the starting input of RNA.

Figure 2 The effect of total RNA concentration on Cq measurements.

(A) The Cq values of four amplicons representing, Beta-Actin, GAPDH, 18S, and P53 were measured using three different total RNA concentrations (50 ng, 100 ng and 200 ng) isolated from 293 HEK cells. (B) Evaluation of Cq values for hsa-miR-21, hsa-miR-99b, Let-7c and U75 at three different RNA concentrations. Using one-way ANOVA to compare the different RNA input groups, there was no significant difference in the Cq values between these groups seen in A and B.

Duplex detection for both mRNA and miRNAs

To further expand upon this utility, we formulated a qPCR reaction to allow parallel detection of both miRNA and RNA transcripts within the same reaction (Table S3). A combination of random primers and miRNA stem loop primers were added into the same reaction mix. RT was performed at the thermal cycling conditions as described in Table S4. For direct comparisons, RNAmp was vetted against the established protocols using random or stem loop primers. A comparison of qPCR output expression of the common reference genes, 18S, Beta-Actin, GAPDH and B2M (Fig. 3A), oncogenes or drivers of cancer progression, CDKN1, p53 and p16 (Fig. 3B) and miRNA biogenesis genes, Dicer and Ago2 (Fig. 3C) was then performed across these 3 RT priming strategies. With RNAmp, there was a decrease of 5 Cq values for the reference genes. p53 and p16 were detected within the shorter cycling timeframe of 3 Cq’s, interpreted as an 8-fold gain. To assess the scope of RNAmp, miRNA machinery genes were additionally examined and detection of the product was increased by four-fold. Importantly, across these gene-sets, RNAmp, when compared to standard approaches did not compromise qPCR detection but instead, enhanced it.

Figure 3 RNAmp increases qPCR sensitivity for mRNA detection.

(A) Comparison of Cq values for reference genes: 18S, Beta-Actin, GAPDH, and B2M using RNAmp versus random priming. (B) Comparison of Cq values for tumour suppressor genes: CDKN1, p53 and p16 using RNAmp versus random priming (C) Comparison of Cq values for Dicer and Ago2 using RNAmp or random priming.

RNAmp was applied to quantify RNA subclasses in a typical siRNA/miRNA knockdown condition using a miRNA and scramble control. Following transfection of VIC labeled hsa-miR-21 or a scramble mimic into HeLa cells, RNA was isolated at 24hrs post transfection and cDNA products generated using RNAmp, random primers and stem loop primers RT protocols. Using the standard methods, hsa-miR-21 could be detected at basal levels in the scramble control and increased in transfected cells. JAG1 is a target of hsa-miR-21 regulation (Hashimi et al., 2009) and a decrease in JAG1 RNA levels was observed in hsa-miR-21 overexpressing cells. This trend was accurately quantified with RNAmp (Fig. 4A). Analysis of the fold change for JAG1 using 2 Delta Cq showed no difference in outcome with either RNAmp or the standard approach (Fig. 4B).

Figure 4 Parallel detection of both miRNA and mRNAs in a single duplex qPCR reaction.

(A) The changes in Cq values for JAG1 and hsa-miR-21 (student t test: **** P < 0.0001, *** P < 0.001). (B) A comparison of JAG1 fold change expressed as 2−ΔCq derived using RNAmp or standard methods. Student t test indicated that the fold change in JAG1 was significant (**** P < 0.0001) and identical to the standard approach. (C) Duplex parallel qPCR detection for hsa-miR-21 and JAG1 in a single volume reaction. For the RNAmp, both qPCR hydrolysis probes for hsa-miR-21 and JAG1 were combined, whereas the RNAmp Singleplex only contained hydrolysis probes for either miR-21 or JAG1. (student t test: **** P < 0.0001, *** P < 0.001). (D) Comparison of RNAmp Duplex versus RNAmp Singleplex for the detection of serum miRNAs.

As RNAmp is a mixture of stem loop and random primers, this setup may introduce amplification bias when compared to single random and stem loop priming methodologies. Thus, qPCR efficiencies were determined for p53, Dicer, and JAG1 using LinRegPCR (Table 2). There was no difference in qPCR efficiencies between RNAmp and standard procedures. This demonstrates that RNAmp exhibits the same qPCR efficiency as random priming methods but provides the parallel detection of different RNA and miRNA species.

Table 2 A comparison of PCR efficiencies between RNAmp versus the standard method for mRNA (Random priming) detection using qPCR.

Method	RNAmp	Random priming	
	JAG1	
Minimum	1.7	1.7	
Maximum	1.8	1.9	
Mean	1.8	1.8	
Std. Deviation	0.067	0.069	
Std. Error of Mean	0.027	0.031	
	p53	
Minimum	1.7	1.6	
Maximum	1.9	2.0	
Mean	1.8	1.8	
Std. Deviation	0.077	0.129	
Std. Error of Mean	0.026	0.043	
	Dicer	
Minimum	1.6	1.8	
Maximum	2.0	1.8	
Mean	1.8	1.8	
Std. Deviation	0.171	0.011	
Std. Error of Mean	0.098	0.006	

Having shown RNAmp as a singleplex qPCR, we evaluated if this approach was applicable in a duplex reaction. Using HeLa samples, hsa-miR-21 (VIC) and JAG1 (FAM) were quantified in a duplex reaction and compared to the standard singleplex qPCR (See Table S5). Both the Cq values and the pattern of gene expression from either the duplex or singleplex RNAmp were similar (Fig. 4C). There was also a shift of 3 Cq’s for JAG1, translating to an 8-fold detection increase. This outcome was also the case for hsa-miR-21, detecting at an earlier cycling frame of 2 Cq’s in the duplex format (4-fold increase in detection).

To expand on the utility of RNAmp for clinical applications the expression of selected miRNAs and mRNA in cancer serum using both the duplex and singleplex qPCR approach was performed (Fig. 4D). In the duplex format, hsa-miR-486-5p, hsa-miR-451 and B2M were all detected in cancer serum. The Cq for the miRNAs were similar in duplex or singleplex format. However, for B2M, the duplex approach offered lower detection limits.

Discussion

We have presented the results of RNAmp; a robust RT to fluorescent qPCR platform, where quantification of miRNAs and mRNAs can occur in parallel.

The observation of smaller volumes providing increase detection at lower Cq has been previously reported (Kamau et al., 2013; Kroh et al., 2010; Lao et al., 2006; Varkonyi-Gasic et al., 2007). However, these studies required a pre-amplification of the RNA, were only applicable to mRNA and limited to a singleplex format. Our approach provides the choice of duplexing different miRNAs alongside mRNAs in small reaction volumes.

A side by side analysis was performed comparing RNAmp to random priming for mRNA detection and stem loop priming for miRNA detection. When compared to these traditional mRNA singleplex reactions, RNAmp did not compromise on detection but rather improved this feature. Nine mRNAs were tested including 18S, Beta-Actin, GAPDH, B2M, CDKN1, p53, p16, Dicer and Ago2. With RNAmp, eight of these mRNAs were detected at a lower Cq, suggesting improved qPCR detection. For B2M, detection was comparable in either qPCR approach. In addition, the detection of miRNAs with RNAmp was similar and comparable with standard methods.

One of the main concerns with RNAmp was the possibility of qPCR efficiency affected by the combination of random and stem loop primers in the cDNA preparation. The efficiency of a qPCR reaction is the increase in amplicons per cycle translating to a value between 1 and 2 (2 being 100% efficiency and doubling of the amplicon per cycle) (Ruijter et al., 2009). The range for most qPCR reactions is between 1.8 and 2.0. Any value below 1.6 is considered a poor performing qPCR reaction. There was no difference in the qPCR efficiency between our approach and standard methods for mRNA and miRNA detection indicating RNAmp is as efficient as these standard procedures. For most mRNAs tested with RNAmp, there was an improve detection (lower Cq), likely due to the small qPCR reaction volumes. The reduction in qPCR reaction “space” at 5.0 µL or 2.5 µL may allow for greater interaction between PCR components, achieving better detection while maintaining qPCR efficiency. Thus, the combination of random and stem loop primers constituting RNAmp, amplify RNA targets up to the same efficiency and in some cases, exceeds the standard protocols. This approach represents improvements to the tried-and-tested hydrolysis chemistry platform widely used amongst researchers.

Another potential consideration of our approach is the limitation of various qPCR instruments to perform the cycling conditions and accurately measure fluorescence at 2.5µL volumes. From the data, qPCR efficiency is not affected by these small volume reactions and the Cq values show minimal variation in the triplicate measurements. We have performed this assay on instruments such as the ABI StepOne Plus, ABI 7500, and ABI 12K Flex with comparable results. The only limitation we have identified is when the reaction volume is set at 1.0 µL, qPCR dynamics and fluorescent measurement are affected at this extreme low volume (Data not shown).

To our knowledge, there is only one other approached which permits the synthesis of miRNA and mRNA species in a single cDNA reaction (miScript SYBR® Green PCR Kit). However, this approach uses SYBR green and does not allow for the detection of both miRNA and mRNAs within a single reaction vessel. Interestingly, there is one report combining in-situ TaqMan PCR with immunostaining to visualize protein expression (Ranjan et al., 2012). This study however, relied on in-situ PCR and the availability of antibodies. With our protocol, the user can (1) simultaneously detect; a miRNA and its mRNA target, (2) the miRNA and a reference gene for normalization and (3) miRNAs and other mRNAs of interest. As a minimum, we have now doubled the qPCR output compared to the standard approaches. Aside from these technical advantages, the ability to perform parallel qPCR analysis of miRNAs and their targets is novel and to our knowledge has not been demonstrated.

The standard volume of a hydrolysis probe assay is 20.0 µL for the detection of a single miRNA/mRNA target. With a qPCR reaction volume between 2.5µL and 5.0µL, this represents a cost reduction of 800%, a benefit when validating RNA biomarkers in large clinical studies. Different RNA targets could be validated at low costs using a single sample thus reducing the usage on clinical samples. In addition, this approach measures two miRNAs simultaneously, again reducing the cost and sample requirements. RNAmp is applicable to serum miRNAs and the validation of multiple serum RNA biomarkers.

RNAmp does limit the user into using Hydrolysis probes as this platform has not yet been demonstrated with SYBR Green based methods. Although probes are commonly used (Bustin & Nolan, 2004), laboratories without this technology would need to commit to an initial investment for this form of chemistry and training. We restricted our testing of RNAmp to the detection of two different RNA families, yet given the number of fluorophores commercially available, it should be possible to measure 12 miRNA/mRNAs/ncRNA in parallel.

Conclusion

In summary, this platform can be used to measure the levels of different miRNAs within a single reaction, demonstrating application both as a tool to measure the miRNA and its regulated target. Moreover, validating multiple miRNAs in serum samples is feasible. PCR efficiency of RNAmp is comparable to standard approaches but RNAmp can also provide improved detection and reduce technical variation as it is a single reaction. When an mRNA target is expressed in low quantities the ability of this platform to provide parallel analysis would reduce both the cost and material requirements. Beyond these benefits, inter-assay variation is negated by the single RNAmp reaction (Bustin et al., 2015). Our approach may be applied for any multi-parametric analysis of gene expression, to measure the role and function of different RNA classes in a single sample.

Supplemental Information

Table S1 RNA Concentration and Quality

Click here for additional data file.

Table S2 First Strand cDNA Synthesis Cycle Conditions

Click here for additional data file.

Table S3 Reaction mix composition of three RT methods

Click here for additional data file.

Table S4 Reaction mix composition of singleplex qPCR reaction

Click here for additional data file.

Table S5 Reaction Mix composition of duplex qPCR reactions

Click here for additional data file.

Table S6 The change in quantification cycle (Cq) values for selected mRNA, miRNAs in smaller fluorescent qPCR reaction volumes

Click here for additional data file.

Table S7 Common Fluorophore bandwidths

Fluorophores with narrow, well-resolved bandwidths are useful for duplex qRT-PCR applications avoiding cross talk. ROX™ is the passive reference utilised in a Step One Plus.

Click here for additional data file.

Supplemental Information 8 Raw Data For Jag1 qPCR

Click here for additional data file.

Supplemental Information 9 Raw data for B2M

Click here for additional data file.

Supplemental Information 10 Raw data for Dicer and Ago2

Click here for additional data file.

Supplemental Information 11 Raw data for B2M and ACTB

Click here for additional data file.

Supplemental Information 12 MIQE checklist and data set

Click here for additional data file.

Supplemental Information 13 Raw data for miR-21

Click here for additional data file.

Supplemental Information 14 Raw data for p53

Click here for additional data file.

Supplemental Information 15 Raw data for Dicer and Ago2 part 2

Click here for additional data file.

We would like to acknowledge Dr Simon Keam, who supplied RNA from the miRNA mimic transfections.

Additional Information and Declarations

Competing Interests

Author Contributions

Human Ethics

Data Availability

The authors declare there are no competing interests.

Samantha Khoury and Nham Tran conceived and designed the experiments, performed the experiments, analyzed the data, prepared figures and/or tables, authored or reviewed drafts of the paper, and approved the final draft.

The following information was supplied relating to ethical approvals (i.e., approving body and any reference numbers):

HREC was granted (UTS HREC 2013000471) by the UTS Human Research Ethics Committee. Written consent for human serum was provided by the patient under this HREC application.

The following information was supplied regarding data availability:

The raw measurements are available as Supplemental Files.

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
