# Peer review of "qPCR multiplex detection of microRNA and messenger RNA in a single reaction"

_PeerJ, doi:10.7717/peerj.9004_

## Round 0.1 · original submission · Minor Revisions

Please take into consideration the reviewer's comments, and provide a revised manuscript and a detailed point-by-point rebuttal letter.

Reviewer 1 ·

Basic reporting

In general, the reading of the article flows, thanks to the quality of writing, the manuscript is clearly written and the length of the article is good. Perhaps, a few words in the scientific context could be improved; for example, the word “patriarch” in line 49, and the word "governing" in line 61, could be changed for other words more correct. Moreover, I advise using keywords that do not appear in the title of the article, to take better advantage of search engine efficiency. Other minor observations, at lines 58 to 71, I noticed a lot abbreviates in the same paragraph that rest clarity to the sentences, I advice to rephrase them.

On the other hand, I think the introduction needs new recent references, the newest reference cited is from 2015, is five years old. The Structure of the paper was according to the PeerJ standards, discipline norm, but can be improved for clarity. I consider that the figures are relevant, have high quality, are well labelled & described. Lastly, I found that raw data supplied is enough and necessary.

Experimental design

It seems to me that the content and theme are congruent with the contents presented in your magazine. The question of investigation, hypotheses and objectives are easily identifiable, although they are not preceded by declaratory phrases such as: "the objective of this investigation was ..."

The validating technique is novel and useful to increase the knowledge of the study area. I could see that they have clarified having made 3 technical replicates per analyzed sample, complying with the sampling requirements to reduce the variability of the data and to confirm their validity. However, I recommend improving the phrases or sentences of lines 144 to 151 to clarify the way in which the experiments were related.

Validity of the findings

I consider that the data is sufficient to validate the findings reported and the speculations made by the authors are mentioned as such in quantity within the permissible limits. All underlying data has been provided; they are robust, statistically sound, & controlled. Conclusions are well stated, linked to original research question & limited to supporting results.

Additional comments

The manuscript is useful in a rising research area within gene expression regulation by small RNAs. One question about the biological samples, do you have informed consentiment forms for all of them? I noticed only one is shown. Why did you named the approach RNAmp? could you describe the meaning at the paper? Did you sequenced the PCR products?

Annotated reviews are not available for download in order to protect the identity of reviewers who chose to remain anonymous.

Reviewer 2 ·

Basic reporting

I commend the authors on the clear and unambiguous, professional English that was used throughout the article. Sufficient background/context was provided and appropriate literature was cited throughout. The article was of professional structure and the raw data and tables provided were clear and transparent. While the hypothesis is very clear, some additional information on how this research is useful to the wider field would be beneficial. Some sections that require modifications or additional clarity are listed below:
Line 31-32: Please consider rephrasing this sentence. A Suggestion: “Our approach termed RNAmp enabled parallel measurement of miRNA and mRNA expression from various cell lines”.
Lines 36-37: Please reconsider the use of the word “sensitivity”. See general comments in Section 4 below.
Lines 58-71: Although you have mentioned that this process requires cDNA synthesis followed by qPCR. I think it is important to highlight that this is a two-step process (i.e. two separate reactions performed in two separate reaction tubes, requiring different reaction components etc.). It would be very convenient if these could be performed together in a single reaction vessel (similar to regular RT-PCR). Is there possibility for this in the future? If not, what are the limiting factors? This could possibly be discussed in the discussion/future work section.
Lines 68-70: You have mentioned one difference between miRNA and mRNA cDNA synthesis (i.e. different primer technologies). Could you please also highlight some of the other differences (e.g. different temperature and enzyme concentration requirements). Do you know why they currently have different requirements?
Lines 72-78: Please explain why it’s important for researchers to detect miRNA and mRNA in parallel? What information does this provide?
Lines 79-87: How do the hydrolysis probes work for miRNA and mRNA? Are the miRNA and mRNA detected using the same probes and in the same fluorescent channel (is it an accumulative signal when miRNA and mRNA are analyzed together?) or are they differentiated? i.e. miRNA and mRNA each produce signal in a different colour channel.

Experimental design

This manuscript is original and is within aims and scope of the journal. The research question is well defined, relevant and meaningful. The authors state how this research fills an identified methodologies gap, however additional information on why researchers use these methodologies would be beneficial. The authors demonstrate rigorous investigation, performed to a high technical & ethical standard. The methods are described with sufficient detail. Some sections that require minor modifications are listed below:
Line 100: “Serum samples were cancer patients and healthy …” Suggestion: “Serum samples were from cancer patients and healthy…”
Line 117-135: My recommendation would be to include this table in the main text and remove the in-text description (Lines 124-135). The in-text description could instead include a broader description about the three cycling conditions and what each of them are used for (please see an example below).
Example for broader in-text description: Three different cycling conditions were performed and are described in Table X. Cycling condition 1 uses stem loop primers for miRNA synthesis as per manufacturer's protocol (Applied Biosystems). Cycling condition 2 uses random primers for coding gene synthesis (i.e. mRNA) as per the manufacturer's protocols (Applied Biosystems). Cycling condition 3 uses a combination of random and stem-loop primers for simultaneous duplex detection of mRNA and miRNA in a single reaction.
Line 138: Please include an introductory sentence to let the readers know why you are testing different reaction volumes.
Line 138: What did you use to dilute the cDNA? Was it dH2O?
Line 140: Again, my recommendation would be to include this table in the main text. These tables are very clear and informative and are easier to follow than the text. You could instead include more generic text about why things are being done. E.g. Why is only 0.8x master mix used for 2.5 uL reaction when 1x master mix is used for all other volumes?
Line 141: Perhaps a table outlining which fluorophore belongs with each target and which targets are duplexed together? I sometimes have trouble following this.
Line 144: Please change “1x volume of Taqman Universal…” to “1x concentration of Taqman Universal…”.
Line 183-191: Please clarify here if Cq was determined using LinReg analysis methods?

Validity of the findings

The authors demonstrate meaningful replication by testing with a range of different mRNA and miRNA biomarkers. The authors have provided all underlying data which are robust, statistically sound, & controlled. The Conclusions are well stated, linked to original research question and are limited to supporting results. I would like to remind the authors that speculation is welcome as long as it is identified as such. Some sections that require minor modifications or additional clarity are listed below:
Lines 197: Please reconsider the use of the wording “increase in qPCR sensitivity”.
Line 206: See above comment
Lines 209-217: Please include a generic statement that highlights this is duplexing two miRNA as opposed to multiplexing miRNA and mRNA (which is discussed later).
Lines 227-235: Can you comment on whether this is an expected finding for qPCR of miRNA and mRNA? Are there any references in the literature to support these findings? Do you have a theory or any speculation as to why this could be? What exact reaction volume was used for these experiments? I can only see that it says “low volumes”.
Lines 237-254: Very interesting. Do you have any theories as to why it was improved using RNAmp? Also, which genes are the miRNA machinery genes?
Lines 248: Wording regarding increase in “sensitivity”. (See general comments below).
Line 250: See comment above.
Lines 360-363: Could you please elaborate on this comment. As far as I am aware, most PCR machines are limited to 5-6 fluorescent channels. Are there other machines with 12 fluorescent channels? Or are you able to detect multiple miRNAs/mRNA/ncRNAs per channel?
Table 1: Perhaps indicate that these PCR efficiencies are for singleplex qPCR reactions.
Table 2: Are these PCR efficiencies for singleplex or multiplex qPCR reactions?
Figure 1: Wording around sensitivity. See general comments. Are the results in Figure 1 singleplex or Multiplex reactions? Please include the amount of RNA added to reactions.
Figure 2A and 2B: Should these graphs say “Reaction Volume” or “Starting template Concentration”? What volume were these reactions performed at? The methods say low volumes, but not exactly which volume.
Figure 4: For the Figure legend, perhaps you could include in brackets which channel was used for JAG-1 and mir-21?

Additional comments

1. Throughout the article you discuss that using a lower reaction volume results in an earlier Cq which suggests increased sensitivity. In PCR, sensitivity represents the smallest amount of substance in a sample that can accurately be measured by an assay. Do you have any data demonstrating that lower reaction volumes improve the limit of detection (LOD)? By this I mean, can you detect lower concentrations of miRNA/mRNA using these lower volumes? If not, my recommendation would be to change the wording of ‘improved sensitivity’ throughout the manuscript. The lower Cq values indicate an earlier or faster detection. However, without testing lower target concentrations and determining the LOD at each volume, I am not sure if it can be assumed that using a lower reaction volume results in better sensitivity.

2. Supplementary Tables 2 - 4 are clear and informative and my recommendation is to include these in the main text. Furthermore, the in-text descriptions could be removed/modified.

3. For the raw data, it looks like technical PCR replicates were performed in triplicate? Could you please confirm this in the methods section?

---

## Round 0.2 · accepted · Accept

The manuscript has improved over the review rounds and it is now accepted at PeerJ.